# Is “Esterhazy II”, an Old Walnut Variety in the Hungarian Gene Bank, the Original Genotype?

**DOI:** 10.3390/plants10050854

**Published:** 2021-04-23

**Authors:** Geza Bujdoso, Benjamin Illes, Virag Varjas, Klara Cseke

**Affiliations:** 1Centre for Horticultural Sciences, Hungarian University of Agriculture and Life Sciences, Park u. 2, 1223 Budapest, Hungary; illesbenjamin15@gmail.com (B.I.); varjas.virag@uni-mate.hu (V.V.); 2Forest Research Institute, University of Sopron, Várkerület 30/A, 9600 Sárvár, Hungary; cseke.klara@uni-sopron.hu

**Keywords:** field observation, old cultivar, Hungary, *Juglans regia*, phenology, SSR markers

## Abstract

The old walnut (*Juglans regia* L.) genotype called “Esterhazy II” was well-known in the Austro-Hungarian Monarchy before World War II, and it can still be found in the Austrian, German and Swiss backyard gardens today. Unfortunately, nowadays, vegetatively propagated progenies of the original “Esterhazy II” are not available anymore around the world because walnut grafting started later than this genotype had become well-known. Although various accessions with “Esterhazy II”-“blood“ are available, it is difficult to determine which one can be considered true or the most similar to the original one. In this paper, phenological and nut morphological characteristics of an “Esterhazy II” specimen planted in a Hungarian gene bank were compared to the varieties “Milotai 10” and “Chandler”. Examined characteristics were: budbreak, blossom time, type of dichogamy, ripening time, nut and kernel features. An additional SSR fingerprinting was used to identify identical genotypes and to demonstrate the relatedness of the analyzed “Esterhazy II” genotype to the other Hungarian walnut cultivars. It can be concluded that under the name “Esterhazy II”, several different genotypes can be observed. All the checked characteristics except budbreak fitted well with the previous descriptions. Our results confirmed that the examined “Esterhazy II” genotype shows high similarity to the “original“ “Esterhazy II” described in the literature.

## 1. Introduction

The basis of the Hungarian walnut assortment is provided by the selected varieties derived from the Carpathian race [1,2], which selections are thought to be native/indigenous [3]. These genotypes have unique characteristics [4], such as early ripening time and large fruit size [5].

Today, the most popular Hungarian bred walnut cultivar is the terminal bearing “Milotai 10” [6]. In the past, some other, old walnut genotypes were the standards, such as “Sebeshelyi gömbölyű”, “Sebeshelyi hosszú”, “Nagybányai” from the territory of Transylvania (Romania) [7], “Milotai” and “Tarpai” from the Hungarian territory [3], and “Franquette” from France [8]. Unfortunately, old cultivars are missing from the contemporary breeding programs of many countries. However there are also some exceptions such as China [9,10], Hungary [2], Iran [11], Kazakhstan [12], New Zealand [13], Romania [14,15], Serbia [16], Slovenia [17] and the United States of America [18]. Apart from the previously mentioned old walnut genotypes, Esterhazy II (syn: Eszterházy II, Eszterházi 2, Estherházai II, E II) still has importance. This genotype was well-known in the territory of the former Austro-Hungarian Monarchy, and it is still popular nowadays, especially in the German-speaking countries. Numerous German, Austrian, and Swiss nurseries sell their “own” genotypes labeled as “Esterhazy II”. However, the very original “Esterhazy II”, being grown from the beginning of the 19th century, is possibly not available anymore since vegetative walnut propagation started later (e.g., in Hungary in the 1970s) than this genotype had become flourishing and widespread. Even if “selected” “Esterhazy II” genotypes can be found in some core collections worldwide, they have a probable seed origin, therefore, may be different from each other.

This paper aims to describe the phenological and fruit morphological characteristics of one “Esterhazy II” specimen, planted in the core collection of the Hungarian University of Agriculture and Life Sciences (HUALS) Research Institute for Fruit Growing and Ornamentals, and to compare it with the two most widespread varieties, “Milotai 1” and “Chandler”. The genotype selected for the study was derived from the initial growing place of the “Esterhazy II” variety from the gene bank in Fertőd. Thus this genotype was considered as a putatively original type. To assess the identity of the various specimens of “Esterhazy II”, compared with the most important walnut varieties in Hungary, genetic fingerprint analysis is conducted, as well. Finally, one of the most important goals of the study is to rediscover and restore this historical variety for the future.

## 2. Results

### 2.1. Phenology

The variety showing the earliest budbreak in the described trial was “Milotai 1”, followed by “Chandler” and “Esterhazy II”. There was a statistical difference at the beginning of budbreak among all the three observed varieties, but concerning the end of budbreak significant difference was found only in the case of “Esterhazy II” (Table 1). The first male catkins of “Chandler” and the first female flowers of “Milotai 1” started to open the earliest, on the same day in the blooming period. “Esterhazy II” was the latest regarding both the first female and first male blooms. “Esterhazy II” and “Chandler” had protandrous dichogamy, but “Milotai 1” had the opposite, protogynous flowering. The beginning and the end of the flowering period, as well as the length of the first male bloom of “Esterhazy II” showed significant differences compared to the standards. However, the beginning and end of the pistillate opening period of “Esterhazy II” were statistically different from “Milotai 1”, but there was no statistical difference between “Chandler” and “Esterhazy II”. Regarding the length of the first female bloom, only “Chandler” differed statistically (Table 2 and Table 3, Appendix A). The harvest time of “Esterhazy II” was between “Milotai 1” and “Chandler”, at the beginning of the third week of September (Table 4).

### 2.2. Nut Characteristics

Nut size is an important feature in the descriptions of walnut varieties and genotypes. Nut height of “Esterhazy II” and “Chandler” were equivalent and differed significantly from “Milotai 10”. The most important characteristic is the nut diameter; “Esterhazy II” reached the significantly largest value among the observed varieties. Again, in the case of nut width, “Esterhazy II” produced the significantly largest value among the examined cultivars (Figure 1).

Varieties can also be distinguished and determined based on the fruit shape. Roundness indices of the observed varieties were very similar, but still, “Esterhazy II” and “Milotai 10” showed significant differences compared to “Chandler”.

Shell thickness values varied between 1.56 and 1.70 mm, where “Esterhazy II” produced the thickest shells; by contrast, “Milotai 10” and “Chandler” had thinner shells and their values differed statistically (Figure 1, Appendix A).

One main aim of walnut breeding is to obtain higher nut production; in other words, many nuts with good nut weight. In our experiment, the dried whole nut weights varied between 11.9 and 14.5 g/nut. The kernel weight was in the range of 5.5 and 6.7 g/kernel. The highest dried nut and kernel weights were measured in the case of “Esterhazy II”; both parameters were significantly higher than the measured values of the other two varieties (Figure 2).

Finally, kernel characteristics determine the successful production of a variety. The ratio of kernel weight compared to the whole fruit weight was in the range of 45.0% and 46.1% for all three varieties. The rate of halves resulted in larger differences (between 73.2% and 86.7%), however. “Chandler” reached the highest value in the category of cracking rate, followed by “Esterhazy II” and “Milotai 10”. There was no significant difference between the varieties examined in this trial regarding this parameter (Figure 3).

Shell and kernel color, especially lightness, is an important feature from the aspect of market value. *L* values of “Chandler” and “Milotai 10” were significantly higher compared to “Esterhazy II”. The “*a*” values of “Milotai 10” were the highest and showed a significant difference from “Esterhazy II”. The “*b*” values of “Chandler” were statistically different from “Esterhazy II” (Figure 4).

### 2.3. Genetic Analysis

Additionally, genetic fingerprint analysis was also conducted since several different specimens are available as possible “Esterhazy II” genotypes besides the analyzed one in this study. However, the genetic uniqueness of these gene bank accessions has never been checked. Our genetic analysis resulted in distinct, unique genotypes in the case of all the selected samples only with one exception, as it was initially suspected. The two “Esterhazy II” specimens from Fertőd are genetically identical, propagated by grafting. Otherwise, under the name “Esterhazy”, several different genotypes can be observed. In general, it can be concluded that the method, with the, applied eight SSR markers, was appropriate for genetic fingerprinting as the P_ID_ value (in other words, the probability of random matching) was very low (1.6 × 10^−5^). All of the eight analyzed SSR markers proved to be polymorphic. The main genetic indices are presented in Appendix A.

The UPGMA dendrogram constructed based on the genetic distance matrix of 18 walnut varieties demonstrates the separation of the analyzed E II (3) specimen from the other genotypes (Figure 5). The trees indicated as E II (1), (2), (3) in the gene bank has a closer relationship with “Alsószentiváni 117” lineages, while the E II and E I trees from Fertőd form a distinct group together with the Hungarian varieties (“Tiszacsécsi 83” and “BD6”) originating from near the Tisza river.

## 3. Discussion

According to the variety descriptions, “Esterhazy II” should have an early budbreak time [19,20,21,22,23,24]. However, former phenology descriptors are not always exact due to the lack of information about the other varieties that had been compared. Nevertheless, it is shown that the “Esterhazy II” genotype, planted in our core collection, has a later budbreak than “Chandler”. There was a significant difference regarding the beginning of budbreak among all the three observed varieties (Table 1), while in the case of other phenology characters, namely the end and length of blooming, the three examined varieties were not statistically different. Hence, in respect of the late budbreak feature of the examined genotype, there is a discrepancy between our observations and the literature data. Descriptions also mentioned that “Esterhazy II” had a protandrous flowering system with long homogamy. As our genotype fits well with these characteristics (Appendix A, Table 2 and Table 3), it can be stated that the genotype may be “Esterhazy II”. Furthermore, the harvest time of “Esterhazy II” (Table 4) also matches the data of the former descriptions [19,20,21,22,23,24].

According to the literature, nuts of “Esterhazy II” have an ovate shape (“egg shape”) with a medium-long tip [19,20,21,22,23,24]. Measurements made in this trial are following this information, as our genotype provided the highest values of nut height and nut diameter (Figure 1 and Appendix A). Nuts from our samples reached the nut height and diameter described in the literature [19,20,21,22,23,24]. Thus, the examined genotype from our gene bank may be true to “Esterhazy II,” also based on the fruit size data.

Regarding shell thickness, “Esterhazy II” should have rather thin shells [21,22,23,24]. Our observations are concordant with these descriptions, as the measured nutshells were significantly thinner compared to the standards involved in this trial. (Appendix A). Furthermore, the shell surface was smooth or slightly grooved, again, similar to what was described in the literature [19,20,21,22,23,24].

The average dried nut weight of “Esterhazy II” from our gene collection varied from 12.7 to 16.3 g/nut. Only one reference [24] mentioned a hint regarding this data, where one kg of dried nuts with shell contained on average 84 nuts. Based on estimations, one kg of dried nuts from the trial should have 62 to 79 nuts/kg. Even if the calculated number of dried nut pieces per kg did not reach the described value, the “Esterhazy II” genotype from our gene bank proved to produce the significantly highest dried nut and kernel weights (Figure 5).

The average kernel rate of the analyzed “Esterhazy II” specimen reached 45.5% (Figure 3), which is a bit lower than the value of 49% mentioned elsewhere [20,21,22,23,24]. Our “Esterhazy II” holds another positive characteristic, which is easy kernel removal, described in the literature [20,21,22,23,24] as well, which resulted in the measured high cracking rate (Figure 3). Kernels of our “Esterhazy II” had a light color, and it differed significantly from “Milotai 10” and “Chandler” also from this aspect (Figure 4). Furthermore, “Esterhazy II” from our collection had an outstanding flavor without bitter aftertaste due to the missing tannin compounds from the kernel tissues [25].

The results of the genetic analysis confirm the previous presumption that the original “Esterhazy II” genotype has various descendants that should be considered as different genotypes in the future, and it would even be worth considering the overwriting of the nomenclature in the case of this variety series. On the other hand, the available “Esterhazy” specimens can serve as a rediscovered gene pool for breeding. For instance, the “Esterhazy II” genotype, planted in our core collection and analyzed here, had been previously selected for a trial due to its remarkable characteristics.

Even if the aim of the genetic analysis presented here was only to check the identity of the various “Esterhazy” specimens, still a cautious inference could be made concerning the origin of the linage. It seems rather probable that the variable holds a considerable part in its gene pool from local ancestors. Therefore, it can stand close to the Carpathian race. However, to reveal the roots of this old walnut cultivar—would it be either a French origin, as it was previously stated in the literature [26], or rather a local, Carpathian lineage—a more detailed genetic analysis, including several reference genotypes from both populations should be accomplished in a future study.

## 4. Materials and Methods

Besides “Esterhazy II”, two other varieties were included in the study. “Milotai 10” is important in the Central European countries, while “Chandler” is among the most grown varieties of the world. By selecting these two varieties, a more comprehensive description can be made regarding several phenological, morphological, and horticultural traits of “Esterhazy II”.

### 4.1. Description of the Varieties Involved in the Trial

#### 4.1.1. “Esterhazy II” Genotype

The Esterhazy II genotype was selected at the beginning of the 19th century from a horticultural plantation in the former Esterhazy estate (today called Fertőd), located in Northwest Hungary, near the Hungarian—Austrian border, and possibly derived from a French origin [26]. The “Esterhazy II” specimen planted in the core collection of the HUALS Research Institute for Fruit Growing was derived directly from Fertőd. The genotype was previously considered as “Esterhazy II”. Therefore, our zero hypothesis is that this genotype represents the type. Henceforth, this genotype is marked under “Esterhazy II” in this paper.

“Esterhazy II” has an early budbreak time; therefore, it is sensitive to the late spring frosts. It has a protandrous flowering; there is long homogamy during its blooming period. The first female bloom date is in early May. The yield of “Esterhazy II” is good or very good in Germany (Internet 4) but poor in Hungarian climatic conditions as it is prone to apomixis. Its harvest time in late September–early October with a solitary setting type. The nuts are ovate with a medium-long tip, large (37 to 45 mm) and approx. 37 mm in width. Its shell is relatively thin, well-closed with a smooth or slightly grooved shell surface. Usually, there are 84 nuts in a kilogram of dried shelled fruits. It is easy to crack, with the kernel rate approx. 49%, and having a unique, so-called ivory white color. Kernel removal is easy and has one of the best flavors. The trees have a medium vigor with wide canopies. This genotype is susceptible to *Xanthomonas arboricole* pv. *juglandis* [19,20,21,22,23,24] (Figure 6).

To check the identity of the variety, all the available specimens under the name “Esterhazy” were sampled for the genetic identification procedure. For this purpose, three additional old trees were sampled from the Esterhazy estate, Fertőd (one “Esterhazy ” and two grafted “Esterhazy II” trees), and three other grafted “Esterhazy II” individuals from the HUALS core collection were added, as well. Furthermore, attempts were made to find any existing herbarium samples of the original “Esterhazy II”, but unsuccessfully.

#### 4.1.2. Milotai 10

Milotai has an early budbreak, with budbreak around late March, early April. Female flowers of this protogynous variety open in the third decade of April; its male flowers shed the pollen in late April, early May. Nuts are ready for harvest around 20 September. “Milotai 10” produces nuts with excellent quality, having a large fruit size, smooth shell surface, light shell and kernel color [5,27] (Figure 6).

#### 4.1.3. Chandler

Chandler is the most widespread hybrid walnut cultivar in the world. In Hungary, its budbreak is medium-early and is mostly a lateral bearer, but this is not typical for the young trees. Nuts ripen in the last week of September. The dried fruits hold high-quality with an average shell weight of 13 g, a diameter of 28–30 mm, are smooth-surfaced, and have a light shell and kernel color. In a well-pruned and irrigated orchard, 90% of the nuts could reach 32 mm in diameter. The kernel rate is 49%. The tree is moderately vigorous, partly upright in habit, but highly susceptible to *Xanthomonas* disease [28,29,30] (Figure 6).

#### 4.1.4. Description of the Trial’s Site Conditions

The trial (47°20′11″ N, 18°51′53″ E, 127 m above sea level) was planted in the spring of 1990, at the experimental field of HUALS Research Institute for Fruit Growing, on chernozem soil with high lime content (pH = 8, total lime content in the top 60 cm layer 5%), and high humus content (2.3–2.5%). Considering the Arany-type cohesion index [31], the K_A_ = 40 refers to medium compactness. All observed trees were grafted on *Juglans regia* selected seedling rootstocks. Three grafted trees were planted in a block of 10 × 10 m. The orchard was not irrigated. During the data collection period, the average annual temperature was 11.5 °C, while the average annual temperature during the growing season (between March and September) was 18.4 °C. The average minimum temperature during the spring months was 5.3 °C, the number of days with frosts during spring (between March and May) was 5.4 days. The average annual precipitation was 580.6 mm, and the sunshine hours were 2079 h/year.

#### 4.1.5. Phenological Observations and Nut Morphology

To collect data concerning budbreak and blossom, the Ctifl scheme [32,33] was used. Starting point (calendar day counted continuously from 1 January every year) for budbreak was the stage of “Cf“ (when the terminal buds reached 2.5 cm length), for the stage of the female flower with the code “Ff2“ (when the stigmas started to open), and for the male flowers “Em“ (when the pollen started to shed). The end of the blossoms was when the male flowers dried and dropped off from the trees, while for the female flowers, the drying out of the feathers marked the end of the blossom. All data were recorded every other or every third day in the mornings between 8 and 11 am.


The harvest time was marked when 50% of the husks were open. The sample of 30 nuts per variety was collected at the harvest time each year between 2010 and 2019, and the nut characteristics (nut height, nut diameter, nut width, shell thickness, dried nut weight, kernel weight) were measured. Roundness index (nut diameter/nut height), kernel rate (weight of kernel/nut weight), and cracking rate (weight of halves/whole kernel weight) were calculated. The data were collected between 2010 and 2019.

Shell and kernel color was measured using a Konica Minolta chromameter CR-400 (Konica Minolta, Japan) color measure, expressing the color with the following three values: “*L*” value is for lightness from black (0) to white (100), “*a*” is from green (-) to red (+), “*b*” is from blue (-) to yellow (+). Measurements were made in the last three years, parallel with the nut examinations.


For statistical evaluation Statgraphics, X64 software was used. The letters a, b, c indicate significantly different groups at SD_5%_, while varieties being not significantly different from each other at SD_5%_ are indicated with the same letter.

#### 4.1.6. Genetic Analysis

SSR fingerprinting was used to check the genetic identity of the “Esterhazy” specimens in the HUALS gene bank. Five “Esterhazy II” samples were included in the analysis indicated as E II (1), E II (2), E II (3) from the HUALS gene bank, E II (Fertőd 1) and E II (Fertőd 2) from Fertőd, supplemented with one additional “Esterhazy I” tree indicated as E I (Fertőd). Sample E II (3) represents the genotype that was the object of the phenology and morphology observations in the field trial.

To gather more information about the “Esterhazy” lineage, the genotype data were also used to demonstrate the relatedness to the Hungarian walnut cultivars. For this purpose, the varieties “Alsószentiváni 117”, “Milotai 10”, “Tiszacsécsi 83”, “BD6” and their hybrids derived from “Pedro” × “Milotai 10” crosses (“Milotai intenzív”, “Milotai bőtermő”, “Milotai kései**^℗^**”), as well as hybrids derived from “Pedro” × “Alsószentiváni 117” crosses (“Alsószentiváni kései**^℗^**”, “Bonifác**^℗^**”) were included in the analysis. Furthermore, “Pedro”, “Chandler”, and “Franquette” were added to represent non-local genotypes in the analysis.

Total genomic DNA was extracted from fresh leaves using a modified procedure of Dumolin et al. 1995 [34]. DNA concentration was measured by spectrophotometry (Eppendorf biophotometer) and standardized to 10 ng/µl. Eight SSR loci were amplified chosen from the literature, namely WGA27, WGA72 [35], WGA89, WGA118, WGA202, WGA276, [36] and JR1817, JR6160 [37]. Each PCR was conducted using a third, universal M13(-21) primer fluorescent-label with 6-FAM, NED, PET or VIC according to the protocol of Schuelke (2000) [38]. Reactions were carried out in a total volume of 15 μL containing 10 ng of template DNA, 1× reaction buffer (GoTaq G2 Flexi, 5× reaction buffer with no magnesium Promega, Madison, WI, USA), 1.5 mM of MgCl_2_ (Promega, Madison, WI, USA), 20 μM of each dNTP (dNTP mix, 10 mM each, Promega, Madison, WI, USA), 1.0 unit polymerase (GoTaq G2 Flexi, 5 U/μL, Promega, Madison, WI, USA), while 0.05 μM of forwarding primer and 0.2 μM of Reverse and M13(-21) primers each (Custom DNA oligos, IDT, Coralville, IA, USA). PCRs were performed in a Veriti Personal Thermocycler (Applied Biosystems, Foster City, CA, USA) with the following steps: an initial denaturation for 5 min at 94 °C, followed by 35 cycles of 45 s at 94 °C, 45 s at the optimum annealing temperature for each marker (T_ANN_ = 55 °C for WGA89, WGA118, T_ANN_ = 58 °C for WGA27, WGA72, WGA202, JR1817, JR6160 and T_ANN_ = 63 °C for WGA276), 60 s at 72 °C. Furthermore, eight additional cycles of 30 s at 94 °C, 45 s at 53 °C (T_ANN_ optimal for M13(-21) primer), 45 s at 72 °C, finished with a final extension step at 72 °C for 10 min. Amplification products were checked on a 2% agarose gel in 1× TAE buffer, stained with GelRed (Biotium, Fremont, CA, USA). Then fragments were diluted (up to 20-fold) for capillary electrophoresis and multiplexed by dye and size in formamide (Hi-Di, Applied Biosystems, Foster City, CA, USA) using GeneScan LIZ 500 (Applied Biosystems, Foster City, CA, USA) internal size standard. SSR genotyping was performed on an ABI 3730 DNA Analyzer (Applied Biosystems, Foster City, CA, USA), while allele calling was carried out using GeneMapper 4.0 software (Applied Biosystems, Foster City, CA, USA).

For SSR data analysis, GenAlEx 6.5 software [39,40] was used. The probability of identity (P_ID_) was calculated for combining the analyzed eight loci by GenAlEx 6.5., possible matching genotypes were checked. The genetic distance matrix was generated by GenAlEx 6.5 for the construction of an UPGMA dendrogram by PAST 4.03 [41] with 9999 bootstrap replicates.

## 5. Conclusions

The old Hungarian walnut genotype labeled as “Esterhazy II” in the HUALS gene bank was examined from phenological and genetical points of view during 2010 and 2019. It can be concluded that under the name “Esterhazy II”, several different genotypes can be observed. Our results confirmed that the examined “Esterhazy II” genotype shows high similarity to the “original” “Esterhazy II” described formerly, since all the checked characteristics fitted well with the literature data, with the only exception of the late budbreak. Nevertheless, only the high similarity can be confirmed since the original “Esterhazy II” is highly probable to have faded away due to the generative propagation used until the 1970s. It was also uncovered that the investigated “Esterhazy II” has a unique genotype. Therefore, it can be considered to be introduced as a new variety from the Esterhazy series. In general, the rediscovery and restoration of this historical variety would provide new candidates for walnut breeding.

## Figures and Tables

**Figure 1 plants-10-00854-f001:**
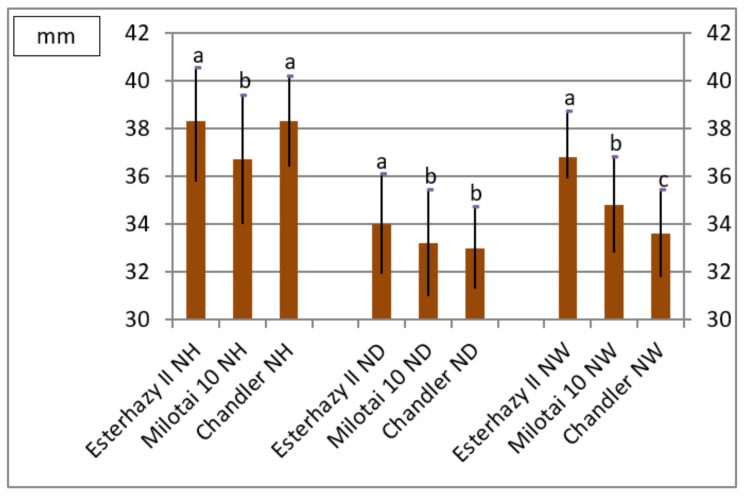
Nut height (NH), nut diameter (ND), and nut width (NW) of the observed varieties (2010–2019) (SD_5%NH_ = 0.9, SD_5% ND_ = 0.8, SD_5%NW_ = 0.7,a,b,c sign significantly different groups at SD_5%_).

**Figure 2 plants-10-00854-f002:**
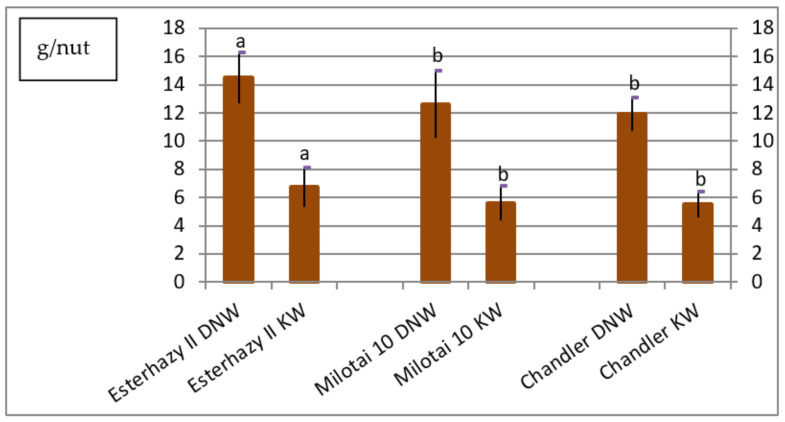
Dried nut weight (DNW) and kernel weight (KW) of Esterhazy II, Milotai 10 and Chandler (2010–2019) (SD_5% DNW_ = 0.7, SD_5%KW_ = 0.5). Varieties being not significantly different from each other at SD_5%_ are indicated with the same letter.

**Figure 3 plants-10-00854-f003:**
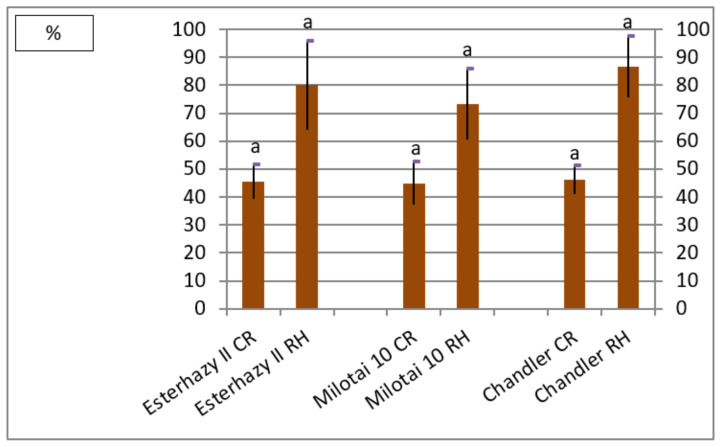
Cracking rate (CR) and the ratio of halves (RH) of Esterhazy II, Milotai 10 and Chandler (2010–2019) (SD_5% CR_ = 2.1, SD_5%RH_ = 14.2). Varieties being not significantly different from each other at SD_5%_ are indicated with the same letter.

**Figure 4 plants-10-00854-f004:**
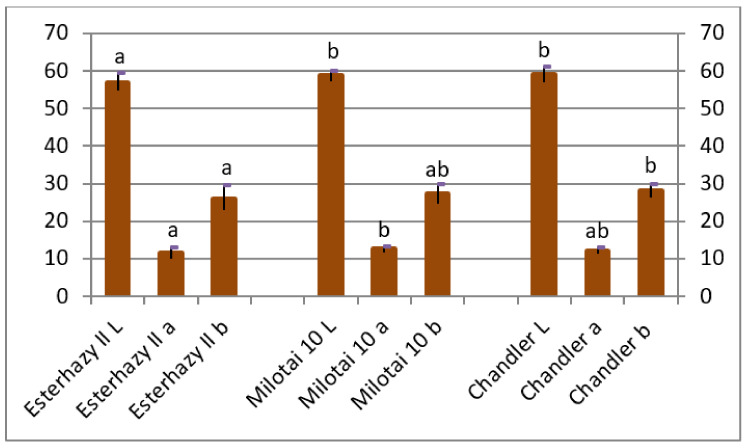
Kernel color of Esterhazy II, Milotai 10, and Chandler using the Lab color system (2010–2019), where the “*L*” value signals lightness from black (0) to white (100), “*a*” from green (-) to red (+), “*b*” from blue (-) to yellow (+). (SD_5%L_ = 1.4, SD_5%a_ = 0.6, SD_5%b_ = 1.5). Varieties being not significantly different from each other at SD_5%_ are indicated with the same letter.

**Figure 5 plants-10-00854-f005:**
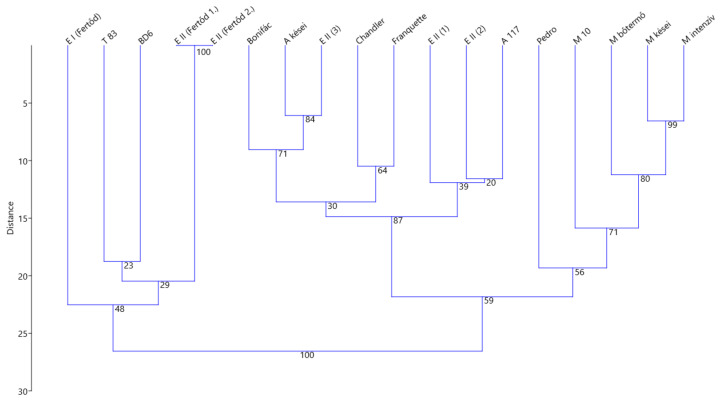
Genetic distance-based UPGMA dendrogram of the selected “Esterhazy” specimens compared with the Hungarian walnut assortment and with three foreign varieties (where A: Alsószentiváni, M: Milotai, T: Tiszacsécsi, E: Esterhazy, with the bootstrap values displayed on the tree at each node).

**Figure 6 plants-10-00854-f006:**
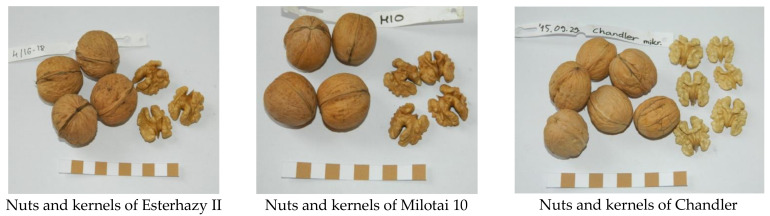
Nuts and kernels of Esterhazy II, Milotai 10 and Chandler.

**Table 1 plants-10-00854-t001:** Beginning, end and length of the budbreak of Esterhazy II, Milotai 10 and Chandler (2010–2019). Varieties being not significantly different from each other at SD_5%_ are indicated with the same letter.

	Esterhazy II	Milotai 10	Chandler	SD_5%_
	Calendar Days (Day, Month) ± SD (Days)			
Beginning of budbreak	27 April ± 3.8 d a	7th April ± 10.6 d bc	16th April ± 10.0 d bc	6.3 d
End of budbreak	4 May ± 4.3 d a	24 April ± 7.3 d b	24 April ± 7.3 d b	5.1 d
Length of budbreak	12.7 d ± 5.2 d a	19.9 d ± 7.5 d ab	19.9 d ± 7.5 d ab	3.8 d

**Table 2 plants-10-00854-t002:** Beginning, end and length of the first male bloom of Esterhazy II, Milotai 10 and Chandler (2010–2019). Varieties being not significantly different from each other at SD5% are indicated with the same letter.

	Esterhazy II	Milotai 10	Chandler	SD_5%_
	Calendar Days (Day, Month) ± SD (Days)			
Beginning of first male bloom	3 May ± 2.8 d a	28 April ± 5.3 d b	23 April ± 4.7 d bc	3.3 d
End of the first male bloom	12 May ± 5.1 d a	7 May ± 3.1 d b	4 May ± 6.4 d ab	4.1 d
Length of first male bloom	9.1 d ± 3.3 d a	9.0 d ± 4.9 d a	11.0 d ± 3.4 d ab	4.4 d

**Table 3 plants-10-00854-t003:** Beginning, end and length of the first female bloom of Esterhazy II, Milotai 10 and Chandler (2010–2019). Varieties being not significantly different from each other at SD5% are indicated with the same letter.

	Esterhazy II	Milotai 10	Chandler	SD_5%_
	Calendar Days (Day, Month) ± SD (Days)			
Start of first female bloom	6 May ± 3.7 d a	23 April ± 7.8 d b	5 May ± 3.7 d a	4.4 d
End of the first female bloom	12 May ± 3.7 d a	5 May ± 5.5 d b	13 May ± 5.1 d a	3.9 d
Length of first female bloom	5.1 d ± 3.3 d a	13.0 d ± 5.7 d a	8.0 d ± 2.3 d b	3.3 d

**Table 4 plants-10-00854-t004:** Harvest time of Esterhazy II, Milotai 10 and Chandler (2010–2019).Varieties being not significantly different from each other at SD_5%_ are indicated with the same letter.

	Harvest Time
	Calendar Days (Day, Month) ± SD (Days)
Esterhazy II	21 September ± 3.2 d a
Milotai 10	17 September ± 6.7 d ab
Chandler	27 September ± 6.4 d a
SD_5%_	5.2 d

## Data Availability

All-new research data were presented in this contribution.

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
