# Peer review of "Is “Esterhazy II”, an Old Walnut Variety in the Hungarian Gene Bank, the Original Genotype?"

_plants, 2021, doi:10.3390/plants10050854_

Round 1
Reviewer 1 Report
Writting should be improved.
Some suggested corrections are shown below:
Line 50. Please add, “and fruit morphological characteristics”
Figure 2. The letters above the bars showing the statistical differences should move close to the bars. Also, authors should add in the Figure legend what the letters mean.
Figure 3. The letters above the bars showing the statistical differences should move close to the bars. Also, authors should add in the Figure legend what the letters mean.
Figure 4. Authors should add in the Figure legend what the letters a,b mean.
Line 171. Please, modify to “Based on the variety descriptions in the literature…”
Line 182. Please delete the words “true to”
Line 183. There is no need to change paragraph.
Line 185. Please add “According to the literature,” in the beginning of the paragraph.
Line 186. Please, change to “Measurements made are….”
Line 194. Please, change to “…grooved, similar to that described….”
Line 288. Please, change to “Collected phenological and nut morphological data”
Paragraph, lines 297-302. Authors should clear describe the way of sampling the 30 nuts taken (e.g. 30 nuts per cultivar? 30 nuts per each year, during the period 2010-2019?).
References
Reference number 20. Please add the year of publication.
Author Response
Reviewer 1
Dear Reviewer 1,
first of all, I take the opportunity to say Thank you for your great work to increase the quality of our work made in the manuscript attached. Your comments are very valuable for all of the authors. Attached I am sending you our comments to your suggested corrections:
Reviewer 1: Line 50. Please add, “and fruit morphological characteristics”
Answer: It was accepted.
Reviewer 1: Figure 2. The letters above the bars showing the statistical differences should move close to the bars. Also, authors should add in the Figure legend what the letters mean.
Answer: It was corrected. Reason of the „mistake” was due to the application of different MS Word programs.
Reviewer1: Figure 3. The letters above the bars showing the statistical differences should move close to the bars. Also, authors should add in the Figure legend what the letters mean.
Answer: It was corrected. Reason of the „mistake” due to the application of different MS Word programs.
Reviewer 1: Figure 4. Authors should add in the Figure legend what the letters a,b mean.
Answer: The meaning of „a”, „b” and „L” is written also in the Materials and Methods (lines 319-320).
Reviewer 1: Line 171. Please, modify to “Based on the variety descriptions in the literature…”
Answer: The marked text was changed for According to the variety descriptions in the literature”.
Reviewer 1: Line 182. Please delete the words “true to”
Answer: The comment was accepted.
Reviewer 1: Line 183. There is no need to change paragraph.
Answer: The comment was accepted.
Reviewer 1: Line 185. Please add “According to the literature,” in the beginning of the paragraph.
Answer: The comment was accepted.
Reviewer 1: Line 186. Please, change to “Measurements made are….”
Answer: The comment was accepted.
Reviewer 1: Line 194. Please, change to “…grooved, similar to that described….”
Answer: The comment was accepted.
Reviewer 1: Line 288. Please, change to “Collected phenological and nut morphological data”
Answer: The comment was accepted. The phenological and nut morphological data were put in a separated subchapter. The name of this chapter was also changed to “Phenological observations and nut morphology” (line 305).
Reviewer 1: Paragraph, lines 297-302. Authors should clear describe the way of sampling the 30 nuts taken (e.g. 30 nuts per cultivar? 30 nuts per each year, during the period 2010-2019?).
Answer: The text was changed to „The sample of 30 nuts per variety was collected at the harvest time each year between 2010 and 2019”.
Reviewer 1: References. Reference number 20. Please add the year of publication.
Answer: The book was published in 2006. This information was added to the reference No. 20.
Finally, the language style was improved in general.
Many thanks for your time and suggestions to improve quality of our manuscript again.
Best wishes,
Geza Bujdoso

Reviewer 2 Report
The manuscript presents an interesting research aiming at characterizing both genetically and phenotypically the remnants of an ancient Hungarian walnut variety, by comparing it with other commonly cultivated varieties. The study was conducted over a very long time, it is overall well-done, and led to the conclusions that, with some exceptions, the extant variety is still very similar to the original one. Such conclusions are relevant, since lay the foundations for future conservation actions of the studied variety.
One major issue regards the style of the English language, that absolutely needs to be much improved. I strongly recommend to commission a revision by a native speaker or at least by an expert.
For the rest, I only have some minor comments.
Title: It would be good to mention Juglans regia here (or walnut, since the scientific name is in the keywords).
Table 2 caption: “…male bloom of…”.
Figures S1, S2, S3: shouldn’t supplements be moved out of the main text?
Figure 1: “Esterhazy II NW” is not fully readable.
Line 191: “Internet 1-4”, wrong way to cite. Please correct here and elsewhere, even in the references (Websites: 9. Title of Site. Available online: URL (accessed on Day Month Year)).
Paragraph 4.1: maybe the descriptive part of the varieties could be moved to the introduction.
Lines 276-282: wasn’t there any risk of spontaneous hybridization between the varieties that could affect already fruit features? Though I guess this didn’t happen, given your results.
Conclusions: This part needs to be enlarged, retracing the highlights of your work. Nevertheless, since it is an optional section, you could either decide to remove it, provided that the two statements that you currently put in the conclusions are moved to the discussion.
Author Response
Reviewer 2
Dear Reviewer 2,
Many thanks for your time and suggestions to improve the quality of our manuscript. Please find our answers here below:
Reviewer 2: One major issue regards the style of the English language, that absolutely needs to be much improved. I strongly recommend to commission a revision by a native speaker or at least by an expert.
Answer: The language style was improved in general, and a native speaker has also checked the manuscript to improve its quality.
Reviewer 2: Title: It would be good to mention Juglans regia here (or walnut, since the scientific name is in the keywords).
Answer: The comment was accepted, the title was change to “Is ‘Esterhazy ll’, an old walnut variety, in the Hungarian gene bank, the original genotype?
Reviewer 2: Table 2 caption: “…male bloom of…”.
Answer: It was changed to „male flowering”.
Reviewer 2: Figures S1, S2, S3: shouldn’t supplements be moved out of the main text?
Answer: The mentioned figures won’t be in the printed version, just on-line, so these will be moved out during the final editing.
Reviewer 2: Figure 1: “Esterhazy II NW” is not fully readable.
Answer: Thank you for this comment, it was corrected.
Reviewer 2: Line 191: “Internet 1-4”, wrong way to cite. Please correct here and elsewhere, even in the references (Websites: 9. Title of Site. Available online: URL (accessed on Day Month Year)).
Answer: It was corrected.
Reviewer 2: Paragraph 4.1: maybe the descriptive part of the varieties could be moved to the introduction.
Answer: In our opinion the varieties’ descriptions are part of the Materials and methods, therefore the authors would like to keep this section here.
Reviewer 2: Lines 276-282: wasn’t there any risk of spontaneous hybridization between the varieties that could affect already fruit features? Though I guess this didn’t happen, given your results.
Answer: Yes, you are right, spontaneous hybridization between the varieties did not happen.
Reviewer 2: Conclusions: This part needs to be enlarged, retracing the highlights of your work. Nevertheless, since it is an optional section, you could either decide to remove it, provided that the two statements that you currently put in the conclusions are moved to the discussion.
Answer: The conclusions part was completed.
Finally, the language style was improved in general.
Many thanks for your time and suggestions to improve the quality of our manuscript again.
Best wishes,
Geza Bujdoso

Reviewer 3 Report
Dear Authors,
From the perspective of the genebank and their work on maintaining and restoring old varieties to cultivation, I think the manuscript topic is very interesting and important. However, it currently has a lot of shortcomings.
First, I don't understand the connection between phenology and genetics. These are two completely separate elements that were forced into one manuscript and caused a disaster.
I did not learn from the manuscript why only the Esterhazy II genotype (3) was used for the phenological and morphological description of nuts and not all 3 found in the HUALS collection. When were individuals obtained for the core collection?
The phenotype section is very decently researched and written. However, the section describing genetic testing is very lacking and has significant errors in inference.
The description of SSR analysis results lacks information about the efficiency of selected markers - i.e. the number of alleles, the level of polymorphism. Due to the small number of samples and I assume a small number of alleles tested analysis of the population structure using STRUCTURE software would not take too much time.
I am extremely worried about the arrangement of the dendrogram. According to it, Chandler and Franquette varieties look like relatively closely related genotypes. A completely different picture of pedigree is presented by the work of :
Nicese, F. P., Hormaza, J. I., & McGranahan, G. H. (1998). Molecular characterization and genetic relatedness among walnut (Juglans regia L.) genotypes based on RAPD markers. Euphytica, 101(2), 199-206.
You, F. M., Deal, K. R., Wang, J., Britton, M. T., Fass, J. N., Lin, D., ... & Dvorak, J. (2012). Genome-wide SNP discovery in walnut with an AGSNP pipeline updated for SNP discovery in allogamous organisms. BMC genomics, 13(1), 1-16.
In contrast, the opposite is true for the Pedro and Chandler cultivars, which, based on this dendrogram, appear to be related to a lesser degree compared to the previously discussed pair. Again, there is an inconsistency with the results of earlier papers.
In my opinion, the resolution of the genetic analysis is too low and the results obtained are biased. It is necessary to increase the number of SSRs or use a second type of molecular markers.
Drawing conclusions based on such results is unauthorized.
In addition, the discussion needs a considerable extension.
Other minor comments are:
the abbreviation HUALS in the title needs to be replaced with either the full name of the institution or the Hungarian core collection. The title also lacks information on what species the manuscript is about. This information appears only in the abstract.
In the description of the methodology, the section on nut characteristics needs to be separated from the phenology.
The results lack dimensioned overview photos of nuts.
I suggest focusing on the analysis of phenotype and phenology and preparing a manuscript on this part only. It should be accompanied by a thorough discussion. Genetics, on the other hand, requires wet analyses. Perhaps after improving laboratory analyses, correct and thorough data analysis it will be material for another publication in the future.
Best regards,
Reviewer
Author Response
Dear Reviewer 3,
First of all, thank you very much for your valuable comments, useful suggestions and for your time in general to improve the quality of our manuscript. Please find our answers here below:
Phenology observations and nut morphology measurements provide the base for the manuscript, while genetic analysis was added in order to check the identity of the different specimens with the same clone name. For this purpose, the application of SSR markers seems to be an appropriate and reliable tool for us. However, we need to agree, that this aim should have been communicated more precisely. Any other conclusions drawn based on the analysis should be treated cautiously since both sample number and marker set were limited at this stage of the research. Therefore, a careful revision of the sections discussing the genetic analysis was made from this aspect. A supplementary table including the main diversity indices (Table S1) was also added. Hopefully, our efforts to elucidate this methodological miscommunication will be appreciated.
Concerning your questions about the other available genotypes:
As it was presented in the manuscript, under the variety name ‘Esterhazy II’ several specimens can be found, however with rather uncertain origin. In our gene bank, for instance, three old individuals can be found (indicated as E II (1), (2), (3)) with the only information that this plant material was originated from Fertőd and they are possibly ‘Esterhazy II’ clones. Please take into consideration, that the genetic identity of these old variety accessions has been never checked before, so this result bears a considerable novelty value.
The reason for why not to present any further phenology and morphology data concerning the two other ‘E II’ trees in the gene bank is very simple: one genotype was selected based on its remarkable characteristics for further analysis 10 years ago and the trial was established with this genotype. The other two genotypes, as well as the accessions located in Fertőd can serve as plant material for similar studies in the future.
Other minor comments were:
Reviewer 3: the abbreviation HUALS in the title needs to be replaced with either the full name of the institution or the Hungarian core collection. The title also lacks information on what species the manuscript is about. This information appears only in the abstract.
Answer: The comment was accepted, the title was changed.
Reviewer 3: In the description of the methodology, the section on nut characteristics needs to be separated from the phenology.
Answer: We accepted the suggestion of Reviewer 2 and we put the phenological and nut morphological data in a separated subchapter. Also the name of this chapter was changed to “Phenological observations and nut morphology” (line 305).
Reviewer 3: The results lack dimensioned overview photos of nuts.
Answer: Three photos were added to the manuscript.
We really hope that the revised manuscript has been improved enough to find it acceptable.
Best regards,
Geza Bujdoso

Round 2
Reviewer 3 Report
Dear Authors,
Unfortunately, there are still significant deficiencies in the revised manuscript. The genetic analysis and interpretation of its results is its weakest point. Its results are seriously biased by the small number of loci tested (8 pairs of SSR primers) and their low polymorphism (only 36 alleles in total). Not much has changed in results analysis either. Unfortunately, I still believe that the manuscript in this form is not suitable for publication.
With best regards
Reviewer.